# Role of Natural Products in the Management of COVID-19: A Saudi Arabian Perspective

**DOI:** 10.3390/healthcare11111584

**Published:** 2023-05-28

**Authors:** Mansour Almuqbil, Sarah Alshaikh, Nura Alrumayh, Fay Alnahdi, Eiman Fallatah, Shahad Almutairi, Mohd Imran, Mehnaz Kamal, Mazen Almehmadi, Ahad Amer Alsaiari, Wafa Ali Abdulrhman Alqarni, Ali Mohammed Alasmari, Sara Alwarthan, Ali A. Rabaan, Moneer E. Almadani, Syed Mohammed Basheeruddin Asdaq

**Affiliations:** 1Department of Clinical Pharmacy, College of Pharmacy, King Saud University, Riyadh 11451, Saudi Arabia; 2Department of Pharmacy Practice, College of Pharmacy, AlMaarefa University, Riyadh 13713, Saudi Arabia; 172220349@student.mcst.edu.sa (S.A.); 172220388@student.mcst.edu.sa (N.A.); 171220283@student.mcst.edu.sa (F.A.); 182220527@student.mcst.edu.sa (E.F.); 171220297@student.mcst.edu.sa (S.A.); 3Department of Pharmaceutical Chemistry, Faculty of Pharmacy, Northern Border University, Rafha 91911, Saudi Arabia; imran.pchem@gmail.com; 4Department of Pharmaceutical Chemistry, College of Pharmacy, Prince Sattam Bin Abdulaziz University, Al-Kharj 11942, Saudi Arabia; m.uddin@psau.edu.sa; 5Department of Clinical Laboratory Sciences, College of Applied Medical Sciences, Taif University, Taif 21944, Saudi Arabia; mazenn@tu.edu.sa (M.A.); ahadamer@tu.edu.sa (A.A.A.); 6Department of Clinical Pharmacy, King Khalid University, Abha 62529, Saudi Arabia; wafaalqarni97@gmail.com; 7Department of Pharmacy, Saudi German Hospital, Khamis Mushet 62217, Saudi Arabia; ali.m.asmari1@gmail.com; 8Department of Internal Medicine, College of Medicine, Imam Abdulrahman Bin Faisal University, Dammam 34212, Saudi Arabia; smalwarthan@iau.edu.sa; 9Molecular Diagnostic Laboratory, Johns Hopkins Aramco Healthcare, Dhahran 31311, Saudi Arabia; arabaan@gmail.com; 10College of Medicine, Alfaisal University, Riyadh 11533, Saudi Arabia; 11Department of Public Health and Nutrition, The University of Haripur, Haripur 22610, Pakistan; 12Department of Clinical Medicine, College of Medicine, AlMaarefa University, Riyadh 13713, Saudi Arabia

**Keywords:** natural products, COVID-19, prevention, treatment, Saudi Arabia

## Abstract

The coronavirus disease of 2019 (COVID-19) pandemic has resulted in an unprecedented circumstance that has never previously occurred. This has caused the Saudi Arabian people to recognize the necessity of preventive measures and explore alternative systems, such as using natural products (NPs), for treating their infection. Therefore, the specific objectives of this study were to explore the factors that influence the selection of NPs for COVID-19 management and to know the outcome of using NPs in COVID-19 infection management. This observational cross-sectional study was conducted in Saudi Arabia between February and April 2022. The validated pretested questionnaire was distributed among different regions of the country via a purposive snowball sampling procedure. Both descriptive statistics and stepwise regression analyses were carried out to evaluate the parameters related to the use of medicinal plants for the prevention of COVID-19 and the treatment of respiratory symptoms during the pandemic. The data obtained were statistically analyzed using IBM SPSS Statistics for Windows, version 25 (IBM Corp., Armonk, NY, USA). Of the 677 participants, 65% reported using NPs for themselves or family members during COVID-19. Utilizing NPs is always given priority by a significant (*p* < 0.001) percentage of survey respondents. Further, a highly significant (*p* < 0.001) percentage of participants felt that using NPs reduced their COVID-19 symptoms without having any remarkable (*p* < 0.001) adverse effects. Family and friends (59%) were the most frequent sources of information about utilizing NPs, followed by personal experience (41%). Honey (62.7%) and ginger (53.8%) were the most utilized NP among participants. Moreover, black seeds, garlic and turmeric were used by 40.5%, 37.7% and 26.3% of the surveyors, respectively. Those who used NPs before COVID-19 were 72.9% more likely to use them during COVID-19. NPs are more likely to be used by 75% of people who live in the central part of the country and whose families prefer it. This is true even if other factors are considered, such as the practice of using NPs along with traditional therapies and the fact that some participants’ families prefer it. Our findings show that NPs were commonly used to treat COVID-19 infection among Saudi Arabian residents. Close friends and family members mainly encouraged the use of NPs. Overall, the use of NPs was high among those who participated in our study; such practices are strongly impacted by society. It is essential to promote extensive studies to improve the recognition and accessibility of these products. Authorities should also educate the people about the benefits and risks of using commonly used NPs, especially those reported in this study.

## 1. Introduction

At the tail end of December 2019, the coronavirus illness, also known as COVID-19 and named after the severe acute respiratory syndrome coronavirus-2 (SARS-CoV-2), was found for the first time in China. The rapid spread of COVID-19 worldwide prompted the World Health Organization (WHO) to declare a public health emergency of worldwide concern [1]. After the first case was discovered in Wuhan, China, COVID-19 quickly spread across all continents, infecting more than 680 million people and claiming the lives of approximately 6.87 million [2]. As a result, the world was coping with the virus’ effects on healthcare, psychological and economic systems. The absence of a prevention strategy led to increased cases, raising the expense of healthcare for hospital stays and palliative therapies. In addition, there was a shortage of diagnostic testing, which contributed to the increased underreporting of cases. Besides fever and coughing, most patients experience chest pain, difficulty breathing or pneumonia. The current gold standard for diagnosing COVID-19 is reverse transcription polymerase chain reaction (RT-PCR), which detects virus RNA in respiratory samples such as nasopharyngeal swabs or bronchial aspirates [3].

SARS-CoV-2 attracted the attention of scientists worldwide who were hoping to discover and develop a treatment for viral infection [4,5]. The most supportive patient care for seriously ill patients included oxygen. Patients in this category were frequently in critical condition and needed respiratory support such as ventilation. Dexamethasone is an example of a corticosteroid that can prolong life in patients with serious and chronic diseases while minimizing the time they depend on a ventilator [6]. The self-prescription of any medication, including antibiotics, as a COVID-19 preventative measure or treatment is not advised by the World Health Organization. Remdesivir was the main antiviral medication the US Food and Drug Administration (FDA) approved for treating hospitalized COVID patients over the age of 12 years [7]. However, the emergence of resistance to Remdesivir has been reported recently [8]. Additionally, two monoclonal antibody medications received the FDA’s emergency use authorization (EUA) to treat COVID-19 [9]. Patients at high risk who have just been diagnosed with mild to moderate disease can receive casirivimab or imdevimab to lower their infection levels and their likelihood of hospitalization [10]. Due to valid concerns about their safety and ability to effectively treat COVID-19 infections, the FDA has revoked its emergency authorization to use hydroxychloroquine and chloroquine to treat COVID-19 patients in hospitals [11]. According to the WHO, vaccination is the best clinical option for successful prevention and control [12]. Apart from this, a significant amount of research is still being carried out to investigate potential therapeutic approaches that are both efficient and risk-free to ease the symptoms of COVID-19 infection. Several studies have focused on exploring the molecular sites that can be targeted to obtain the desired results more effectively. A study described the spike protein’s (S-protein) receptor-binding domain’s (RBD) interaction with human cells via the host angiotensin-converting enzyme 2 (ACE2) [13]. Another study contends that vapreotide may be suitable for preventing SARS-CoV-2 infection [14]. Although the COVID-19 infection rate is decreasing, the emergence of several mutations has caused global concerns [15]. As a result, developing strategies or procedures for effectively controlling this infection is an absolute necessity. 

The term “natural products” (NPs) refers to a complex mixture of organic substances derived from any part of the plant, such as the leaves, stems, flowers, roots and seeds [16]. The bioactive constituents extracted using a solvent are responsible for most NPs’ therapeutic effects [17]. NPs have contributed significantly to drug discovery, particularly for cancer and infectious disorders [18,19], as well as being a common source of therapeutic compounds for treating cardiovascular disease (such as statins) and multiple sclerosis (such as fingolimod) [20,21]. NPs have distinct characteristics and have undergone structural “optimization” by evolution to carry out particular biological activities. These activities include interacting with other organisms and controlling innate defensive mechanisms. Generally, substances regarded as “bioactive” are more abundant in the NPs pool. Compared to conventional synthetic small-molecule libraries, these molecules occupy a wider area of the chemical space [22].

The release of inflammatory cytokines in COVID-19 patients causes a cytokine storm and immunological dysregulation, resulting in acute respiratory distress syndrome and multiorgan failure [23,24]. Therefore, adopting NPs may promote immunity and guard against adverse consequences [25,26,27]. The available research supports using NPs against SARS-CoV-2 due to their antiviral, anti-inflammatory, antioxidant, immune-supportive and preventative activities [28]. NPs control innate and adaptive immunity in a multidirectional manner, regulating immune function [29].

The use of NPs can contribute immensely to the development of novel antiviral medicines with improved efficacy and a better safety index. Current research on several naturally occurring compounds’ antiviral mechanisms has shown how these substances can work with the viral life cycle, specifically with viral entrance, replication, assembly and release, as well as how these mechanisms target virus–host interactions specific to each virus [30].

The concept that NPs can assist in treating and preventing epidemic diseases has been strengthened by several studies employing NPs to treat the SARS coronavirus (SARS-CoV) [31]. During the COVID-19 pandemic, the global use of NPs expanded considerably and the findings are encouraging [32,33]. By limiting SARS-CoV-2 proliferation and entrance into host cells, NPs can affect COVID-19 pathology. *Citrus* spp., orange (*C. sinensis*), *Allium sativum*, *Allium cepa*, *Mentha piperita* and *Nigella sativa* are among the most sought-after antiviral medicinal plant species that can provide helpful adjuvant components in COVID-19 management [34]. Many natural substances, including cordycepin, gallinamide A, plitidepsin, telocinobufagin and tylophorine, have undergone extensive research to treat COVID-19 infection [35,36].

Some recent research from Saudi Arabia has been conducted on using NPs [37,38]. Still, there is no update on the reasons that motivated the selection of NPs and these studies also do not evaluate the impacts of their use. Further, several researchers have explored the use of NPs only among the urban population [39]; therefore, this study was performed to investigate the use of NPs among the Saudi Arabian population from different regions of the country, the factors that influence their selection and the consequences of their use.

## 2. Materials and Methods

### 2.1. Study Design, Participants and Settings

This observational cross-sectional study was carried out in Saudi Arabia between February and April 2022. Participants in the study were Saudi residents who were at least 18 years old and had voluntarily agreed to participate. Using a method known as purposive snowball sampling, the validated and pretested questionnaire was sent out to be completed by respondents from various locations around the country. The participants were given links to online questionnaires developed using Google Forms. The study’s aims and the terms of informed consent were outlined at the beginning of the online form. Participants were allowed to either participate in the study or decline, making their participation a choice. The participant was requested to register their response by self-administration. The research proposal was approved by the institutional ethical committee of AlMaarefa University with reference number IRB06-0603022-20 (dated 6 March 2022).

### 2.2. Determination of Sample Size

As of 2022, the total population of Riyadh city is estimated to be 36,722,767, as reported by the world population review (https://worldpopulationreview.com/countries/saudi-arabia-population, accessed on 30 January 2023). Therefore, our study’s sample size was 385 based on the online sample size calculator http://www.raosoft.com/samplesize.html (accessed on 12 February 2022), keeping a 5% margin of error and a 95% confidence level.

### 2.3. Study Questionnaire, Validation and Pretest

The research team developed the questionnaire with the help of the published literature. Further, it was validated with the help of experts in community health, epidemiology, infectious medicines, social health and pharmacy practice professionals. The questionnaire was translated into Arabic with the help of bilingual professionals using the forward and backward methods. As part of the pilot/pretest, a questionnaire was initially distributed to 30 eligible participants to determine if there needed to be more understanding of any of the study questions. Some questions and statements were rephrased at the end of the pilot study to improve its understanding. Since some changes were made to the questionnaire at the end of the pilot study, these responses were excluded from the final analysis. The reliability of the study questionnaire was confirmed by checking the Alpha Cronbach factor, which was found to be 0.81. Finally, a bilingual (Arabic and English) questionnaire was used for the study. 

### 2.4. Study Questionnaire

There were four sections in the questionnaire used in the study (Appendix A). The participants had to complete all sections and items included in each section. The four sections were sociodemographic characteristics, COVID-19 infection status in the family, use of NPs in treating diseases and use of NPs in treating COVID-19.

#### 2.4.1. Sociodemographic Characteristics

This section had eleven items to determine the age of the participants, their nationality, gender, marital status, educational level, income range, employment status, geographical location, the status of their chronic ailments, the status of their usage of any prescription drugs and commitment to MOH instruction regarding COVID-19.

#### 2.4.2. COVID-19 Infection Status

This section explored the status of COVID-19 infection and its severity level (mild, moderate, severe). In addition, it was meant to determine the vaccination status (yes or no) of the participants, including the number of jabs (one, two or three doses). 

#### 2.4.3. Use of NPs in the Treatment of Diseases

This section was developed to extract information on the practice of using NPs by the participants before COVID-19 (always, never, sometimes), the practice of combining NPs with prescription drugs (yes, no) and the practice of using NPs to improve immunity (yes, no). In addition, it recorded the sources that recommended the use of NPs for the participants, such as the internet and social media, relatives/friend and health practitioners, based on previous experience, by reading books and any other sources.

#### 2.4.4. Use of NPs for the Treatment of COVID-19 

This section was meant to determine the participants’ practice of using NPs for treating COVID-19 (always, never, sometimes), NP use by the family for COVID-19 management (yes, no), the status of relief from symptoms after using NPs (yes, no) and side effects if they experienced any after using NPs (yes, no). In addition, this section explored the commonly used NPs by the participants to treat COVID-19, such as honey (obtained from Apis mellifera), ginger (*Zingiber officinale*), black seed (*Nigella sativa*), garlic (*Allium sativum*), turmeric (*Curcuma longa*), cinnamon (*Cinnamomum verum*), Indian installment (*Withania somnifera*), myrrh, cumin (*Commiphora myrrha* Nees L.), chamomile (*Matricaria chamomilla* L.), clove (*Syzygium aromaticum)*, star anise (*Illicium verum*), lemon grass (*Cymbopogon citratus*, Stap), thyme (*Thymus vulgaris* L.) and any other. Participants could select one or more NPs they used regularly or include their choice if not included in the list. Further, participants were also asked about the most common symptoms for which they used NPs; they were given a list of symptoms that included coughs, sore throat, tiredness, fever, headache, loss of taste and smell, abdominal pain, aches and pains, diarrhea and any other symptom.

### 2.5. Data Analysis

The data collected were entered into IBM SPSS Statistics for Windows, version 25 (IBM Corp., Armonk, NY, USA). Univariate descriptive analysis of the socio-demographic characteristics of the study sample using the Pearson Chi-square test was conducted. The factors influencing the decision to use NPs were determined using stepwise regression analysis to calculate the odds ratio (significant coefficient). A *p*-value of less than 0.05 was significant. The influence of independent variables on dependent variables was determined using stepwise regression analysis. This resolves multicollinearity and predicts the likelihood of the influence of the independent variable in the presence of additional variables on the dependent variable. It also excludes those variables that may not contribute to changing the dimension of the dependent variable.

In our analysis, using NPs to manage COVID-19 infection was kept as the dependent variable and all other factors were considered independent variables. The independent variables were age (below 40 years vs. above 40 years), regions (central vs. other regions of location in Saudi Arabia), educational level (less than university vs. university education), employment status (employed vs. unemployed), marital status (married vs. single/separated), income level (below SAR 10,000 vs. above SAR 10,000), level of commitment to MOH guidelines (committed vs. noncommitted), the severity of COVID-19 infection (mild vs. moderate to severe), hospitalization status (hospitalized vs. home stay), vaccination status (yes vs. no), doses of vaccination (up to two doses vs. three doses), frequency of NP use (regular vs. irregular), NP use by family (yes vs. no), NP use before COVID-19 infection (yes vs. no), frequency of NP use before COVID-19 (yes vs. no), the priority of NP use (yes vs. no), relief after using NPs (yes vs. no), NP side effects (yes vs. no), chronic disease (yes vs. no) and on prescription medications (yes or no).

## 3. Results

### 3.1. Sociodemographic Characteristics of the Participants

Among the 677 participants in this study, 72% were female (Appendix A). Three fourths (75%) of the study sample were Saudi nationals and 50% were participants aged 26–40. Most surveyors were from Saudi Arabia’s central region (55%), with 23%, 9%, 8% and 6% from the western, northern, eastern and southern regions, respectively. Furthermore, 62% of the participants were undergraduates and 54% were employed. Almost half (48%) of those who took part had a monthly family income of more than SAR 10,000. About 57% of the respondents were married and more than 18% had chronic illnesses for which they were taking medications. Almost 60% of participants said they are completely committed to the Ministry of Health (MOH) instructions for COVID-19.

### 3.2. Sociodemographic Characteristics with the Use of NPs for Disease Management

As shown in Table 1, a significantly (*p* = 0.048) high percentage of participants in all age groups agreed that they either constantly or occasionally utilize NPs in their daily routine to treat various ailments. On the use of NPs, female participants significantly outnumbered male participants (*p* = 0.013). A significant (*p* < 0.001) proportion of employed and unemployed participants use NPs; however, unemployed people use NPs more frequently than employed individuals. Further, a significantly large number of married individuals prefer using NPs compared to single individuals (*p* = 0.008). No significant association was noticed between the use of prescription drugs and the use of NPs. The correlation between the prevalence of chronic diseases and the use of NPs was also insignificant. Many of our study samples reported the use of NPs irrespective of whether they were Saudi nationals or expatriates working in the country. There was no significant association between the use of NPs and their income levels. The geographical location of the participants also did not significantly influence the use of NPs. Finally, the commitment to MOH guidelines had no significant impact on the use of NPs.

### 3.3. Sociodemographic Characteristics and the Use of NPs for COVID-19

As exhibited by Table 2, a significantly (*p* = 0.033) large number of participants, especially those above the age of 26, agreed that they used NPs for the treatment of COVID-19. Further, a significantly high proportion of unemployed (0.021) and those who are married (0.002) have acknowledged greater use of NPs when compared to other participants. 

### 3.4. The Use of NPs for COVID-19 and Their Practices

Among the study samples, the use of NPs was significantly (*p* < 0.001) high among those participants who had mild to moderate levels of COVID-19 infection (Table 3). A significant (*p* < 0.001) number of participants used NP at home. A significantly (*p* < 0.001) high percentage of participants who used NPs during COVID-19 also had a practice of using them before COVID-19 infection; many of them used NPs in combination with conventional drugs and used them as an immunity enhancer. 

### 3.5. The Use of NPs during COVID-19 and Its Outcomes

A significant (*p* < 0.001) number of surveyors in this study always prioritize using NPs (Table 4). In addition, a significant (*p* < 0.001) number of family members practice using NPs to treat COVID-19 infection. Further, a significant (*p* < 0.001) number of surveyors believe that they were relieved from the symptoms of COVID-19 after using NPs without any remarkable (*p* < 0.001) side effects. 

### 3.6. Sources of Information for the Use of NPs

Figure 1 shows that relatives and friends (59%) were the most common source of knowledge for using NPs, followed by their previous experience (41%). Furthermore, 19%, 12% and 11% of the surveyors cited social media, health practitioners and book reading as sources of information for using natural goods, respectively.

### 3.7. Distribution of Most Utilized NPs for COVID-19

Honey (62.7%) and ginger (953.8%) were the two most widely used NPs among our survey participants. Furthermore, 40.5%, 37.7% and 26.3% of the surveyors utilized black seeds, garlic and turmeric, respectively. Other NPs used by our study samples included cinnamon, Indian installment, myrrh, cumin, chamomile, clove, star anise, lemongrass and thyme (Figure 2).

### 3.8. Distribution of COVID-19 Symptoms for Which NPs Are Used

Around 65% of the participants of this study acknowledged the use of NPs either for the treatment or prevention of COVID-19 symptoms. Half of the study sample used NPs to treat cough symptoms in COVID-19, while sore throat (46%) was the second most common reason for using NPs among the study sample. The usage of NPs was influenced by fatigue and fever, which accounted for 33% and 31% of the time, respectively. Headache, loss of taste, abdominal pain and general body aches/pain were cited as reasons for using natural items in COVID-19 therapy by one fifth of the study participants. Diarrhea was given as a reason for using NPs by 12% of the sample (Figure 3). In addition, honey, ginger and garlic were selected by 42, 37 and 25% of study participants for addressing COVID-19 infection-related cough symptoms. Additionally, it was found that 40.17, 35.59 and 25.55% of the study sample mentioned utilizing honey, ginger and black seeds, respectively, to treat sore throats brought on by COVID-19. Honey, garlic and black pepper were employed by 28.06%, 23.19% and 17.57% of the survey participants to combat COVID-19-related fatigue, respectively.

### 3.9. Stepwise Regression Analysis

The influence of independent variables on dependent variables was determined using stepwise regression analysis. Stepwise regression resolves multicollinearity and predicts the likelihood of influence of the independent variable in the presence of additional variables on the dependent variable. It also excludes those variables that may not contribute to changing the dimension of the dependent variable.

In our analysis, the use of NPs for managing COVID-19 infection was kept as the dependent variable and all other factors were considered independent variables (Appendix A). The independent variables were age (below 40 years vs. above 40 years), regions (central vs. other regions of location in Saudi Arabia), educational level (less than university vs. university education), employment status (employed vs. unemployed), marital status (married vs. single/separated), income level (below SAR 10,000 vs. above SAR 10,000), level of commitment to MOH guidelines (committed vs. noncommitted), the severity of COVID-19 infection (mild vs. moderate to severe), hospitalization status (hospitalized vs. home stay), vaccination status (yes vs. no), doses of vaccination (up to two doses vs. three doses), frequency of NP use (regular vs. irregular), NP use by family (yes vs. no), NP use before COVID-19 infection (yes vs. no), frequency of NP use before COVID-19 (yes vs. no), priority of NP use (yes vs. no), relief after using NPs (yes vs. no), NP side effects (yes vs. no), chronic disease (yes vs. no) and on prescription medications (yes or no).

Table 5 (model 5) below explains the coefficients of five regression analysis models. The fifth model was taken for interpretation of the analysis. The beta coefficients are all significant and in a logical direction. The use of the NPs by the participants before COVID-19 infection was the strongest predictor for the use of NPs during COVID-19 infection. Further, the participants’ attitude to prioritizing the use of NPs for all purposes was the second predictor for the use of NPs for treating COVID-19. Subsequently, the routine use of NPs by the family for managing other chronic diseases, the practice of combining with conventional drugs and belonging to the central region of Saudi Arabia were other predictors of the use of NPs during COVID-19 infection. 

Appendix A summarizes the overall impact of all significant factors. The first significant factor produces a 72.9% change in the dependent variable; that is, the chance of using NPs increases by 72.9% among those using NPs before COVID-19. This will increase to 74.2% (74.2 − 72.9 = 1.3%), with an additional 1.3% among those routinely prioritizing NP use. Further, this will increase to 74.6% in people who practice adding NPs to conventional therapies. Additionally, this will rise to 74.8% in those whose family practices using NPs. Moreover, in the final step, we find an increase of 75% among those who live in the central region of Saudi Arabia.

## 4. Discussion

This study was conducted to determine the level of NP use among Saudi Arabian people and the possible motivating factors for that decision. A total of 65% of the participants acknowledged using NPs for themselves or their families as part of COVID-19 treatment.

In 2021, a Saudi Arabian study [40] found that 64% of participants utilized NPs as a preventative treatment to boost their immune systems during the peak of COVID-19. Our study further supports their findings because a sizable majority of the 677 participants—64.79%—acknowledged using NPs at the time of COVID-19 infection, whether they were the ones affected or when their family members were infected. In another study [38] that looked at the number of NPs used among a randomly selected sample of 809 Saudi adults before the COVID-19 pandemic, 88.4% of the participants agreed to have used NPs at some point in their lives, with many of them utilizing these supplements for therapeutic purposes (88.7%). It is, therefore, commonly known that Saudi Arabian people employ NPs to treat their diseases. During the COVID-19 pandemic, it is likely that the system of using NPs remained the same or perhaps expanded. Alyami et al.’s [39] analysis revealed that just 22.1% of NPs were used during the outbreak, contrasting our findings and the conclusions of the two papers described above. This was far less than what we found and what was previously reported in the literature. One of the reasons could be that the study was short-lived and collected data during a lockdown. In contrast, our data collection period lasted approximately three months and included information from several facets of society. Moreover, we gathered data considerably later than the pandemic’s most likely peak phases. Our study participants had more time to be influenced by their friends, family, the media, news channels and other resources.

The majority of our study’s participants (59%) were influenced by their relatives and friends when choosing NP, while 41% of the surveyors chose NPs based on their prior use of NPs. A total of 19% of people say that the internet and social media impact their choice of NPs. An earlier study [39] conducted in Saudi Arabia found that the internet and social media had a far higher influence (39.4%), whereas family and friends only had a 14.7% influence. It is probable that the internet and media impacted the participants to a greater extent in the earlier study than their own personal experiences, family and friends because interaction with family and friends was less common during the height of COVID-19 in October 2020.

Most study participants chose NPs when they experienced respiratory problems. In 46% of the surveyors, a sore throat caused them to use NPs first, whereas 50% began using it when they had a cough. These results are similar to those of a study conducted in Peru [41], where it was shown that respiratory symptoms were the primary trigger for patients to consider using NPs. Studies have been carried out on the use of NPs during the COVID-19 pandemic by various societies and cultures worldwide, particularly in Asian nations, including India, China, Japan and Pakistan, as well as some regions of Africa [42]. Inflammation and hemotoxicity are associated with COVID-19 symptoms. Hence, blood-purifying NPs with anti-inflammatory, antioxidant and antiviral activities may be suitable for COVID-19 treatment [43]. In our study, we found that most participants (62.7%) managed COVID-19 using honey alone or in combination with other NPs and traditional substances. At least 181 compounds in honey contribute to its various pharmacological effects [44]. It may function directly by inhibiting SARS-CoV-2 or inadvertently by enhancing the immune response.

With a usage rate of 53.8%, ginger was the second-most popular natural substance among the participants in our study. It has a wide variety of pharmacological effects due to the large number of biologically active compounds it contains, making it useful in the treatment of many diseases and infections. A number of investigations on ginger’s possible therapeutic benefits in treating infections and respiratory ailments [45] have been conducted [46]. Given that COVID-19 is an infectious respiratory condition, it will likely have a major positive impact.

For more than 2000 years, black seed (*N. sativa*) has been used in traditional and prophetic Arabic NPs, particularly in the Middle East, to treat various illnesses, including skin conditions, asthma, coughs, bronchitis, headaches, fevers and influenza [47]. More than 40% of the study participants preferred black seed as a COVID-19 therapy option. More than 37% of the participants preferred garlic as another natural substance. The health benefits of garlic include immunomodulatory, antimicrobial, antioxidant, anti-inflammatory, anticarcinogenic, antihypertensive, antithrombotic, antidiabetic and prebiotic properties. Garlic’s secondary metabolites can be separated into active molecules with and without sulfur. Allicin and alliin are the primary sulfur compounds, whereas flavonoids and saponins are the main sulfur-free active chemicals [48]. Due to the production of hydrogen bonds between amino acids in the binding site of the major structural protease of SARS-CoV-2 and garlic’s bioactive components, which is the protease responsible for viral replication, garlic has been shown to have the ability to suppress SARS-CoV-2 [49]. Patients with COVID-19 typically have higher levels of leptin, TNF-∝, interleukin-1, interleukin-2 and interferon-gamma but lower levels of helper T cells, cytotoxic T cells, regulatory T cells and natural killer (NK) cells (IFN-∝). Notably, garlic consumption significantly lowers levels of leptin, leptin receptor, TNF-∝, IL-6 and proliferator-activated receptor gamma while significantly raising levels of helper T cells, cytotoxic T cells and NK cells (PPAR-). Due to its capacity to influence cytokine release, immunoglobulin synthesis, phagocytosis and macrophage activation, garlic may also be used as a COVID-19 therapy [50].

The regression analysis used in this study indicated a link between the patients’ prior usage of NPs for routine purposes and their use of NPs for COVID-19 treatment. Given that the Saudi community supports NPs for treating ailments, this likely impacted the extensive usage of NPs during the pandemic [39]. In addition, we found that the central location of Saudi Arabia and the priority of NP usage are all related to NP use and the ease with which they can be mixed with other medicines.

The population of Saudi Arabia’s central area is primarily urban, which includes the capital city, Riyadh. The surveyors from the Riyadh region indicated that they use NPs more regularly. Individuals who live in cities are more likely to be well-educated and to react quickly to changing circumstances. The fact that urban residents know it reflects how most of the country’s population is perceived. Because of this, individuals in this area consume NPs significantly more frequently than people in other parts of the nation. The widespread use of NPs in Saudi Arabia is comparable to that in other nations. The National Health Commission in China has authorized the combination of NPs and conventional drugs as a COVID-19 treatment [32]. According to a Cochrane systematic review, NPs taken with modern therapy may help SARS-CoV patients with their symptoms and quality of life [51]. A meta-analysis that included seven randomized controlled trials that assessed the effectiveness of NPs in resolving COVID-19 symptoms, such as fever and dry cough, found that using a combination of NPs with prescription medications resulted in a quicker recovery from these symptoms [34]. Therefore, combining NPs with modern medicines for managing COVID-19 has positive implications.

Although NPs are mostly considered safe and devoid of major adverse effects, there is, however, a possibility of minimizing or magnifying the therapeutic effects of conventional drugs when combined [52]. Therefore, numerous educational campaigns should be launched to inform the public about the potential herb–drug interactions [53] and promote the safe use of therapeutic regimens with or without adding NPs.

In our study sample, 83.33% and 64.30% of COVID-19 nonvaccinated and vaccinated samples, respectively, used NPs, which was statistically not significant. Nevertheless, this is an additional indicator of a high proportion of the use of NPs among the study participants. During the peak period of COVID-19, several researchers explored the role of NPs as a source to develop vaccines. In comparison to the lengthier period of months required for cell culture-based approaches [54,55], NPs provide an effective platform for the production and manufacture of biological products on a wide scale within a few weeks. Therefore, NP-based research should be explored more extensively to improve the productivity of biological preparations. We did acknowledge several limitations even though we were effective in figuring out the frequency of NP usage and the factors that might have influenced the choice to employ them. This study cannot precisely identify the relationship between research variables due to the limited sample size and cross-sectional study design. Online surveys used to gather data may have led to inaccurate reporting or misunderstanding of the questions’ elements. However, we took considerable measures to speak with the respondents directly and clear up any misunderstandings. The uneven geographic distribution of the participants is another issue. Even though more than 40% of the population lives in the country’s center, it would have been advantageous to have more participants from other regions to present our findings more convincingly as the results of a national study. There is a possibility of recall bias in this study because it involves the recall of the participant’s experience with using NPs. Infection with COVID-19, on the other hand, is considered an event and remembering events is easier than remembering experiences. Furthermore, considering that most of the study’s participants were under 40 years old and educated, we assumed that the recall bias had little to no influence.

## 5. Conclusions

The number of people in our survey who used NPs was quite high; such behaviors are significantly influenced by the society in which they are present. Further, the use of NPs was especially high among those who had the practice of using NPs before the beginning of the pandemic. Many participants used honey alone or along with other NPs; this was followed by ginger, black seed and garlic. It is crucial to encourage thorough research, especially when these NPs are combined with other NPs or conventional medicines, so that appropriate actions may be taken to facilitate the approval process by the authorities and greater acceptability by society. 

## Figures and Tables

**Figure 1 healthcare-11-01584-f001:**
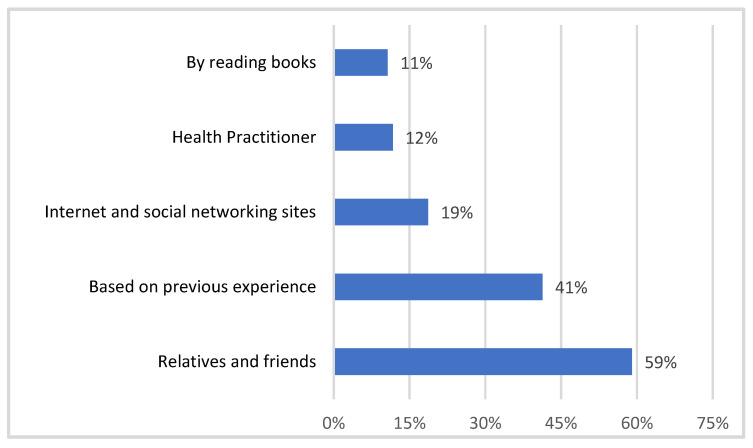
Sources of information on the use of NPs, *n* = 677.

**Figure 2 healthcare-11-01584-f002:**
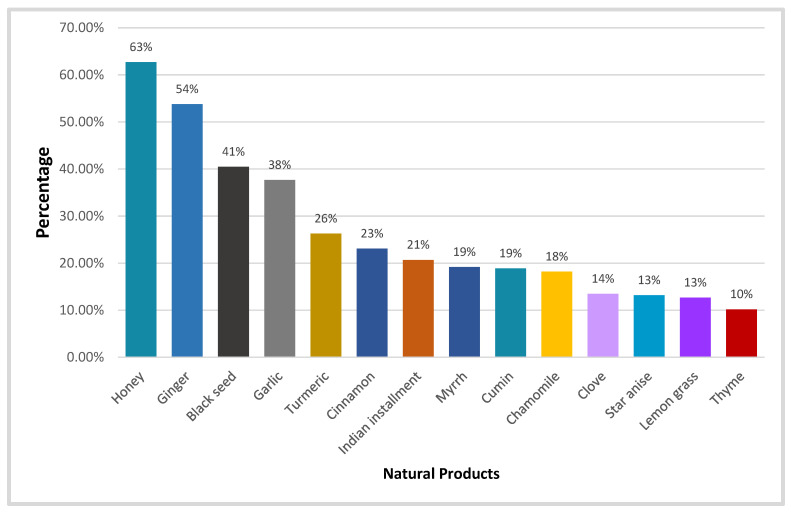
The percentage distribution of most utilized NPs for COVID-19, *n* = 677.

**Figure 3 healthcare-11-01584-f003:**
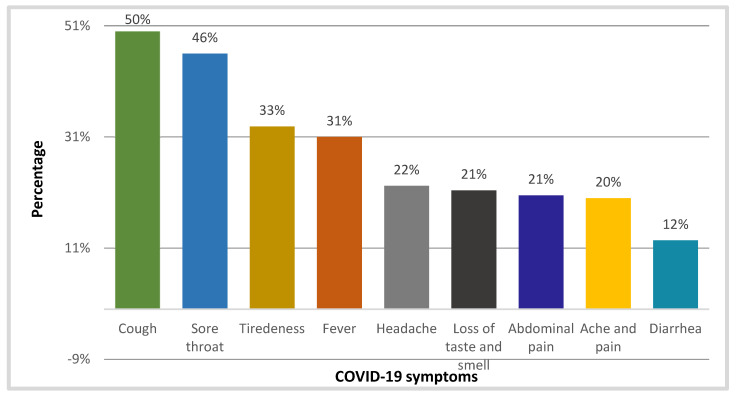
Percentage distribution of COVID-19 symptoms for which NPs were used *n* = 677.

**Table 1 healthcare-11-01584-t001:** Comparison of sociodemographic characteristics with the use of NPs for disease management.

Variables	Options	Use of NPs for Disease Management, [*n* (%)]			Total (677)	*p* Value *
		Always	Sometime	Never		
Age	18–25	52 (29)	99 (55)	28 (16)	179	0.048
	26–40	128 (38)	166 (49)	42 (13)	336	
	More than 40 years	55 (34)	95 (59)	12 (7)	162	
Gender	Female	182 (37)	254 (52)	50 (10)	486	0.013
	Male	53 (28)	106 (55)	32 (17)	191	
Nationality	Saudi	178 (35)	270 (53)	57 (11)	505	0.519
	Non-Saudi	57 (33)	90 (52)	25 (14)	172	
Geographical location	Central region	115 (31)	208 (56)	48 (13)	371	0.069
	Eastern region	26 (50)	18 (35)	8 (15)	52	
	Northern region	17 (29)	32 (55)	9 (15)	58	
	Southern region	18 (45)	19 (47)	3 (7)	40	
	Western region	59 (38)	83 (53)	14 (9)	156	
Educational level	Intermediate school	5 (18)	21 (75)	2 (7)	28	0.061
	High school diploma	48 (29)	96 (57)	23 (14)	167	
	Undergraduate	159 (38)	213 (51)	46 (11)	418	
	Postgraduate degree	23 (36)	30 (47)	11 (17)	64	
Employment status	Employed	123 (34)	197 (54)	45 (12)	365	0.001
	Not employed	100 (45)	92 (42)	28 (13)	220	
	Student	12 (13)	71 (77)	9 (10)	92	
Marital status	Married	146 (38)	199 (52)	38 (10)	383	0.008
	Single	73 (28)	149 (57)	41 (16)	263	
	Separated	16 (52)	12 (39)	3 (10)	31	
Family income	Less than SAR 5000	51 (35)	67 (46)	27 (19)	145	0.060
	SAR 5000–10,000	74 (36)	112 (55)	18 (9)	204	
	More than SAR 10,000	110 (33)	181 (55)	37 (11)	328	
Chronic diseases status	Yes	35 (29)	69 (57)	18 (15)	122	0.255
	No	200 (36)	291 (52)	64 (11)	555	
Use of prescription drugs	Yes	31 (26)	68 (58)	19 (16)	118	0.069
	No	204 (36)	292 (52)	63 (11)	559	
Commitment to MOH instructions for COVID-19	Fully commitment	133 (33)	218 (54)	54 (13)	405	0.658
	Committed to some extent	97 (38)	135 (52)	27 (10)	259	
	Not committed at all	5 (39)	7 (54)	1 (8)	13	

* *p* value by using Pearson’s chi-squared test by comparison of use of NPs for disease management with the variables given in the first column.

**Table 2 healthcare-11-01584-t002:** Comparison of sociodemographic characteristics with the use of NPs for COVID-19.

Variables	Options	NP Use Status for COVID-19 [*n* (%)]			Total (677)	*p* Value *
		Yes	No	Not Applicable		
Age	18–25	66 (37)	57 (32)	56 (31)	179	0.033
	26–40	152 (45)	73 (22)	111 (33)	336	
	More than 40 years	82 (51)	33 (20)	47 (29)	162	
Gender	Female	223 (46)	113 (23)	150 (31)	486	0.417
	Male	77 (40)	50 (26)	64 (34)	191	
Nationality	Saudi	217 (43)	126 (25)	162 (32)	505	0.454
	Non-Saudi	83 (48)	37 (22)	52 (30)	172	
Geographical location	Central region	163 (44)	89 (24)	119 (32)	371	0.083
	Eastern region	19 (37)	13 (25)	20 (39)	52	
	Northern region	23 (40)	23 (40)	12 (21)	58	
	Southern region	18 (45)	6 (15)	16 (40)	40	
	Western region	77 (49)	32 (21)	47 (30)	156	
Educational level	Intermediate school	14 (50)	4(14)	10(36)	28	0.136
	High school diploma	63(38)	47 (28)	57 (34)	167	
	Undergraduate	201 (48)	95 (23)	122 (29)	418	
	Postgraduate degree	22(34)	17 (27)	25 (39)	64	
Employment status	Employed	160 (44)	84 (23)	121 (33)	365	0.021
	Not employed	110 (50)	46 (21)	64 (29)	220	
	Student	30 (33)	33 (36)	29 (32)	92	
Marital status	Married	186 (49)	79 (21)	118 (31)	383	0.002
	Single	94 (36)	78 (30)	91 (35)	263	
	Separated	20 (65)	6 (19)	5 (16)	31	
Family income	Less than SAR 5000	64 (44)	30 (21)	51 (35)	145	0.608
	SAR 5000–10,000	96 (47)	50 (25)	58 (28)	204	
	More than SAR 10,000	140 (43)	83 (25)	105 (32)	328	
Chronic diseases status	Yes	51 (42)	32 (26)	39 (32)	122	0.777
	No	249 (45)	131 (24)	175 (32)	555	
Use of prescription drugs	Yes	48 (41)	29 (25)	41 (35)	118	0.642
	No	252 (45)	134 (24)	173 (31)	559	
Commitment to MOH instructions for COVID-19	Fully commitment	181 (45)	89(22)	135(33)	405	0.558
	Committed to some extent	114 (44)	70(27)	75 (29)	259	
	Not committed at all	5 (38)	4(31)	4 (31)	13	

* *p* value by using Pearson’s chi-squared test comparing NP use status for COVID-19 with the variables given in the first column.

**Table 3 healthcare-11-01584-t003:** Comparison of the use of NPs for COVID-19 with their practices.

Variables	NP Use Status for COVID-19 [*n* (%)]				*p* Value *
	Yes	No	Not Applicable	Total (677)	
Vaccination status					0.307
Yes	290 (44)	161 (24)	210 (32)	661	
No	10 (62)	2 (12)	4 (25)	16	
Severity of COVID-19					0.001
Mild	144 (70)	51 (25)	10 (50)	205	
Moderate	92 (72)	31 (24)	4 (3)	127	
Severe	6 (60)	3 (30)	1 (10)	10	
Hospitalization/home					0.001
Hospital	8 (57)	5 (36)	1 (7)	14	
Home	238 (71)	84 (25)	14 (4)	336	
NP use before COVID-19					0.001
Always	140 (60)	33 (14)	60 (26)	233	
Never	11 (14)	36 (46)	31 (40)	78	
Sometimes	149 (43)	93 (27)	107 (31)	349	
Combination of NP with other drugs					0.001
Yes	243 (52)	89 (19)	138 (29)	470	
No	57 (27)	74 (36)	76 (37)	207	
Use of NP for immunity					0.001
Yes	259 (50)	94 (18)	161 (31)	514	
No	41 (25)	69 (42)	35 (32)	163	

* *p* value by using Pearson’s chi-squared test comparing NP use status for COVID-19 with the variables given in the first column.

**Table 4 healthcare-11-01584-t004:** Comparison of NP use during COVID-19 with outcomes.

Variables	NP Use Status for COVID-19 [*n* (%)]			Total (677)	*p* Value *
	Yes	No	Not Applicable		
Priority for NP use					
Always	115 (60)	21 (11)	56 (29)	192	0.001
Never	28 (24)	51 (44)	37 (32)	116	
Sometimes	157 (43)	91 (25)	121 (33)	369	
NP use by family for COVID-19					0.001
Yes	284 (64)	58 (13)	99 (22)	441	
No	16 (7)	105 (45)	115 (49)	236	
Status of relief from symptoms					0.001
Yes	272 (65)	68 (16)	78 (19)	418	
No	14 (32)	29 (66)	1 (2)	44	
Side effects					0.001
Yes	8 (53)	3 (20)	4 (27)	15	
No	280 (62)	101 (22)	72 (16)	453	

* *p* value by using Pearson’s chi-squared test comparing NP use status for COVID-19 with the variables given in the first column.

**Table 5 healthcare-11-01584-t005:** Coefficients of five models of regression analysis (Model 5).

**Coefficients ^a^**										
**Model**		**Unstandardized Coefficients**		**Standardized Coefficients**	**t**	**Sig.**	**95% Confidence Interval for B**		**Collinearity Statistics**	
		**B**	**Std. Error**	**Beta**			**Lower Bound**	**Upper Bound**	**Tolerance**	**VIF**
5	(Constant)	0.267	0.037		7.294	0.001	0.195	0.339		
	NP use before COVID	0.497	0.014	0.790	36.399	0.001	0.470	0.524	0.787	1.271
	Priority for NP use always	0.098	0.019	0.106	5.243	0.001	0.061	0.135	0.898	1.113
	Combination of NPs with other drugs COVID-19	0.041	0.015	0.054	2.706	0.007	0.011	0.071	0.922	1.084
	NP use by family	0.059	0.021	0.059	2.741	0.006	0.017	0.100	0.799	1.251
	Regions of Saudi Arabia	0.032	0.013	0.046	2.393	0.017	0.006	0.059	0.996	1.004

^a^ Dependent variable: NPs for COVID-19.

## Data Availability

Data is contained within the article or Appendix A.

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
