# Peer review of "Role of Natural Products in the Management of COVID-19: A Saudi Arabian Perspective"

_healthcare, 2023, doi:10.3390/healthcare11111584_

Round 1
Reviewer 1 Report
The study examined the use of natural products as a management of COVID-19. The manuscript is well written.
Specific comments:
- The authors sent a pre-tested questionnaire to respondents from various locations around the country. However, it is unclear if the 30 participants who completed the pre-test survey were excluded in the post-survey. It would be helpful if the authors could discuss this in the method and discussion sections.
- The authors utilized stepwise regression, but they did not specify the type of regression used. If they used logistic regression, it would be beneficial for the authors to report the odds ratio in Table 5.
- The author presented the results of the natural product usage and the factors that influenced it and reported that most participants received the vaccine. Additionally, those who consumed natural products had mild, moderate, or severe COVID-19 outcomes and were more likely to be hospitalized. To provide readers with a complete picture, the authors should conduct an analysis examining the relationship between natural product usage and COVID-19-related outcomes.
- The authors should discuss the impact of using both vaccines and natural products in the discussion section.
- The authors should discuss the impact of recall bias in the discussion section.
Author Response
Reviewer 1
Comments and Suggestions for Authors
The study examined the use of natural products as a management of COVID-19. The manuscript is well written.
Specific comments:
- The authors sent a pre-tested questionnaire to respondents from various locations around the country. However, it is unclear if the 30 participants who completed the pre-test survey were excluded in the post-survey. It would be helpful if the authors could discuss this in the method and discussion sections.
Response: Thank you so much for your important suggestion. We have excluded the responses of 30 participants in the final analysis. We have included a statement on this in the methods section.
- The authors utilized stepwise regression, but they did not specify the type of regression used. If they used logistic regression, it would be beneficial for the authors to report the odds ratio in Table 5.
Response: Thanks for your valuable suggestion. However, we would like to inform the respected reviewer that we have used a linear stepwise regression model that demonstrates the influence of each independent variable on the dependent variable in a step-by-step manner. In Table 5, we demonstrate the best-fit Model 5 that has been explained below the table 5 with reference to supplementary Table 3 as follows:
Supplementary Table 3 summarizes the overall impact of all significant factors. The first significant factor produces a 72.9% change in the dependent variable; that is, the chance of using NP increases by 72.9% among those using NP before COVID-19. This will further increase to 74.2% (74.2-72.9=1.3%), with an addition of 1.3% among those who routinely prioritized NP use. Further, this will increase to 74.6% in people who practice adding NP to conventional therapies. Additionally, this will rise to 74.8% in those whose family practices using NP. Moreover, in the final step, we find an increase of 75% among those who live in the central region of Saudi Arabia.
- The author presented the results of the natural product usage and the factors that influenced it and reported that most participants received the vaccine. Additionally, those who consumed natural products had mild, moderate, or severe COVID-19 outcomes and were more likely to be hospitalized. To provide readers with a complete picture, the authors should conduct an analysis examining the relationship between natural product usage and COVID-19-related outcomes.
Response: Thanks for the comment.
We have already analyzed the relationship between the use of NP and COVID outcomes using a Chi-square non-parametric test where we found that there was a significantly increased number of participants who reported for recovery from symptoms after using NP. This has been given in Table 4 and explained in lines 274-276.
- The authors should discuss the impact of using both vaccines and natural products in the discussion section.
Response: Unfortunately, among our study participants, we did not find any relationship between the use of natural products and the use of NP. This has been explained in Table 3. Further, we have included a description of this relationship in the discussion section.
- The authors should discuss the impact of recall bias in the discussion section.
Response: Thanks for your suggestion. We have now included the impact of recall bias and its mitigation in the discussion section.
There is a possibility of recall bias in this study because it involves the recall of the participant's experience with the usage of NP. Infection with COVID-19, on the other hand, is considered an event, and remembering events is easier than remembering experiences. Furthermore, considering that most of the study's participants were under 40 years old and educated, we assumed that the recall bias had little to no influence.5. Conclusions

Reviewer 2 Report
Title; Role of Natural Products in the Management of COVID-19: A Saudi Arabian Perspective
Comments;In my view, the results obtained in this study are worthy for publication. The manuscript needs major essential revision before publication. I would like to overview the revised version of the manuscript. I have the following comments/suggestions for authors to address before final decision on the manuscript.
1. How can natural products be integrated with conventional medical treatments for COVID-19?
2. What are some of the most promising natural products that have been studied for COVID-19 management in Saudi Arabia?
3. What are the potential risks associated with the use of natural products in COVID-19 management?
4. What is the current state of research on natural products and their effectiveness in managing COVID-19?
5. Are there any ethical concerns regarding the study design, data collection, or reporting of results?
6. Authors have suggested to specify the gaps in the literature review or areas where present research is conducted?
7. Authors have suggested to provide additional insights or recommendations that could strengthen the study's findings and contribute to the field's understanding of natural products' role in managing COVID-19.
8. In the Introduction section the author should refer to the research paper and comment on recent in-silico techniques. It will be good information for the readers. I would like to recommend several papers, among many others, providing further explanation on this topic:PMID: 34346317 PMID: 32399096
9. In the MS authors have highlighted the Role of Natural Products in the Management of COVID-19 specially in the Saudi Arabian region. There are several experimental and computational studies from across the world suggesting potential application of natural products in management of COVID-19. Authors did not provide any valuable insights from this study.
10. “Those who used NP before COVID-19 were 72.9% more likely to use it during COVID-19.” What message do authors are trying to convey from this? It is not understood.
11. "Despite the fact that the participants reported no notable side effects, it would be desirable to do an evidence-based study on the NPs that are often used to ensure their safety." Authors have reviewed traditionally used NP using back from centuries. They are already regarded as safe. IT is a very illogical statement made by the authors.
12.. Conclusion lacking the future aspects and major findings.
13. The study conducted by the authors is quite generalized. To produce more favorable outcomes for their objectives, the authors ought to include an extensive and substantial statistical analysis.
Minor editing of English language required
Author Response
Reviewer 2
Comments and Suggestions for Authors
Title; Role of Natural Products in the Management of COVID-19: A Saudi Arabian Perspective
Comments;In my view, the results obtained in this study are worthy for publication. The manuscript needs major essential revision before publication. I would like to overview the revised version of the manuscript.
Response: Thank you very much for taking out time to review this manuscript and sharing valuable comments.
I have the following comments/suggestions for authors to address before final decision on the manuscript.
1. How can natural products be integrated with conventional medical treatments for COVID-19?
Response: We have included a detailed discussion on the possibility of combining NP with modern medicines and its merits in treating COVID-19 in discussion section from line 428 to 437.
What are some of the most promising natural products that have been studied for COVID-19 management in Saudi Arabia?
Response: As per our study, honey was most commonly used NP that is followed by ginger, black seed, garlic, turmeric, cinnamon and so on. The details are available in figure 2.
- What are the potential risks associated with the use of natural products in COVID-19 management?
Response: Out of the 601 participants who used NP for managing COVID-19, only 8 reported mild nature of side effects such as wheezing with honey, abdominal discomfort with ginger, and diarrhea with garlic.
What is the current state of research on natural products and their effectiveness in managing COVID-19?
Response: We have already discussed the current status of the effectiveness of NP in COVID-19 in the introduction section. Further, we have included the following statement.
Many natural substances, including cordycepin, gallinamide A, plitidepsin, telocinobufagin, and tylophorine, have undergone extensive research to treat COVID-19 infection [20].
Furthermore, more details about the use of NP along with conventional drugs are included in discussion section as part of comment 1 response.
Are there any ethical concerns regarding the study design, data collection, or reporting of results?
Response: Thanks for your comment. We obtained the study’s ethical approval and included an informed consent form in the questionnaire.
The details are given in ‘Study design, participants, and settings’ subsection under Materials and Methods section.
Authors have suggested to specify the gaps in the literature review or areas where present research is conducted?
Response: We have included the gap in the literature in the last paragraph of the introduction that we tried to fill in with our research.
Authors have suggested to provide additional insights or recommendations that could strengthen the study's findings and contribute to the field's understanding of natural products' role in managing COVID-19.
Response: We have included several recommendations in the discussion section now.
Although NP is mostly considered safe and devoid of major adverse effects, however, there is a possibility of minimizing or magnifying the therapeutic effects of conventional drugs when combined [40]. Therefore, numerous educational campaigns should be launched to inform the public about the potential herb-drug interactions [41] and promote the safe use of therapeutic regimens with or without the addition of NP.
Therefore, NP-based research should be explored more extensively to improve the productivity of biological preparations.
In the Introduction section the author should refer to the research paper and comment on recent in-silico techniques. It will be good information for the readers. I would like to recommend several papers, among many others, providing further explanation on this topic:PMID: 34346317 PMID: 32399096
Response: We are thankful to the reviewer for the suggestion. Now we have included both references.
In the MS authors have highlighted the Role of Natural Products in the Management of COVID-19 specially in the Saudi Arabian region. There are several experimental and computational studies from across the world suggesting potential application of natural products in management of COVID-19. Authors did not provide any valuable insights from this study.
Response: We are thankful for the comment of the respected reviewer; we have included additional references for the use of NP in COVID-19.
We hope that we have covered all necessary areas in the introduction section that can provide a solid foundation for the study and help readers understand the direction of our research.
“Those who used NP before COVID-19 were 72.9% more likely to use it during COVID-19.” What message do the authors are trying to convey from this? It is not understood.
Response: we are trying to explain that those who have a practice of using NP to manage their regular ailments will have a probability of using NP for treating COVID-19, the proportion of 72.9 times out of 100.
"Despite the fact that the participants reported no notable side effects, it would be desirable to do an evidence-based study on the NPs that are often used to ensure their safety." Authors have reviewed traditionally used NP using back from centuries. They are already regarded as safe. IT is a very illogical statement made by the authors.
Response: We regret this statement and are now corrected in the revised version.
12.. Conclusion lacking the future aspects and major findings.
Response: Thanks for your comments. Included now.
The study conducted by the authors is quite generalized. To produce more favorable outcomes for their objectives, the authors ought to include an extensive and substantial statistical analysis.
Response: We have done extensive statistical analysis; there are three tables with more in-depth analysis included in the supplementary file. Hope it addresses the concern of the respected reviewer.

Reviewer 3 Report
Manuscript “Role of Natural Products in the Management of COVID-19: A Saudi Arabian Perspective” is informative manuscript. Below are some comments/suggestion for the authors:
- Language should be improvised.
- In Introduction section, there are many facts written but are not cited. Write something on novelty/importance of this manuscript in this section.
- Several reports available on role of natural products are not covered, viz. 10.3389/fphar.2022.926507; 10.1039/D1RA00644D; 10.1016/j.sajb.2021.03.012; 10.1002/ptr.6873. Each facts should be supported with the reference. Role of natural products in COVID complications and COVID variants (10.1016/j.envres.2021.112240) may also be discussed somewhere in manuscript.
- Methodology section should contain inclusion and exclusion criteria.
- In methodology section under heading “Use of NP for the treatment of COVID-19”, it is not clearly mentioned that list of NP is provided by the author to participant or participants there self fill the name of NP. Please clearly state it.
- Provide the Latin names of NP mentioned in Figure 2.
- Discuss more the possible reason involved in difference of use of NP among different sociodemographic areas.
- Also author can provide the data about use of which NP is for what symptom.
- Which NP is used for the highest found symptom. For eg For Cough
- This data should be evaluated more with the help of statistician to find associations between all these different parameters. So, that it can provide more leads.
Language and grammar should be improved
Author Response
Reviewer 3
Comments and Suggestions for Authors
Manuscript “Role of Natural Products in the Management of COVID-19: A Saudi Arabian Perspective” is informative manuscript. Below are some comments/suggestion for the authors:
- Language should be improvised.
Response: Thanks for taking out time to review our manuscript. We have extensive language editing throughout the manuscript.
- In Introduction section, there are many facts written but are not cited. Write something on novelty/importance of this manuscript in this section.
Response: Thanks for the comments, we have included some more details in the introduction section and added the number of references to strengthen our statements. Hope this will satisfy the respected reviewer.
- Several reports available on role of natural products are not covered, viz. 10.3389/fphar.2022.926507; 10.1039/D1RA00644D; 10.1016/j.sajb.2021.03.012; 10.1002/ptr.6873. Each facts should be supported with the reference. Role of natural products in COVID complications and COVID variants (10.1016/j.envres.2021.112240) may also be discussed somewhere in manuscript.
Response: Reference number 1 (10.3389/fphar.2022.926507) is included in the manuscript.
And also we have included 10.1016/j.envres.2021.112240. Zhao Y, Huang J, Zhang L, Chen S, Gao J, Jiao H. The global transmission of new coronavirus variants. Environmental research. 2022 Apr 15;206:112240.
- Methodology section should contain inclusion and exclusion criteria.
Response: The inclusion criteria is already available in section methods under
subsection ‘Study design, participants, and settings’
‘Participants in the study were Saudi residents who were at least 18 years old and had voluntarily agreed to participate.
- In methodology section under heading “Use of NP for the treatment of COVID-19”, it is not clearly mentioned that list of NP is provided by the author to participant or participants there self fill the name of NP. Please clearly state it.
Response: A standard list of NPs was provided to the participants. However, there was a choice given to the participants if they were using NPs that were not included in the list, they could name them.
- Provide the Latin names of NP mentioned in Figure 2.
Response: The latin names of the NPs mentioned in Figure 2 are given in 2. Materials and Methods- subsection, Use of NP for the treatment of COVID-19
honey (obtained from Apis mellifera), ginger (Zingiber officinale), black seed (Nigella sativa), garlic (Allium sativum), turmeric (Curcuma longa), cinnamon (Cinnamomum verum), Indian installment (Withania somnifera), myrrh, cumin (Commiphora myrrha Nees L.), chamomile (Matricaria chamomilla L.), clove (Syzygium aromaticum), star anise (Illicium verum), lemon grass (Cymbopogon citratus, Stap), thyme (Thymus vulgaris L.
- Discuss more the possible reason involved in difference of use of NP among different sociodemographic areas.
Response: More details are included now under sub section Sociodemographic characteristics with the use of NP for disease management
- Also author can provide the data about use of which NP is for what symptom.
Response: Thanks for your suggestion, we have included a description of the use of different NPs among the top three symptoms of COVID-19 under the sub-section Distribution of COVID-19 symptoms for which NP is used.
- Which NP is used for the highest found symptom. For eg For Cough
Response: It’s included now in the the sub-section Distribution of COVID-19 symptoms for which NP is used.
- This data should be evaluated more with the help of statistician to find associations between all these different parameters. So, that it can provide more leads.
Response: Thanks for the suggestion. We have improved the evaluation of the data with the help of statisticians. Three more tables are available in the supplementary file to support the findings of this study. Hope this will satisfy the respected reviewer.

Reviewer 4 Report
1. Check the abbreviations throughout the manuscript and introduce the abbreviation when the full word appears the first time in the abstract and the remaining for the text and then use only the abbreviation (For example, COVID-19, TNF-∝, IL-6, etc.,). Make a word abbreviated in the article that is repeated at least three times in the text, not all words to be abbreviated. The usage of abbreviations in the keywords may be avoided.
2. If possible, in introduction, the authors may cite recent prevalence or incidence data since the author have cited March 2023 data and it may be April 2023 for better understanding for the needs of this investigation.
3. When referring to SPSS versions beginning from 19, authors should cite ‘IBM SPSS Statistics for Windows, version 25 (IBM Corp., Armonk, N.Y., USA)'.
4. The authors should mention properly the word, “TNF-∝” instead of “TNF-”, all over the manuscript.
5. The conclusion seems very general. All conclusions must be convincing statements on what was found to be novel, impact based on the strong support of the data/results/discussion. Moreover, the authors may also be included the limitation of the present findings for a better understanding of the manuscript.
1. The English need improvement since there are some grammatical and syntax errors in the manuscript. For example,
· in line number 114, the word “better” may be as “a better”;
· in line number 175, “Also it” as “Also, it”;
· in line number 176, “two ,or” as “two, or”;
· in line number 387, “Number” as “A number”.
Author Response
Reviewer 4
Comments and Suggestions for Authors
- Check the abbreviations throughout the manuscript and introduce the abbreviation when the full word appears the first time in the abstract and the remaining for the text and then use only the abbreviation (For example, COVID-19, TNF-∝, IL-6, etc.,). Make a word abbreviated in the article that is repeated at least three times in the text, not all words to be abbreviated. The usage of abbreviations in the keywords may be avoided.
Response: Thank you very much for your comments. We have prepared the abbreviation section and included it at the end of the manuscript before references. Only the abbreviated keyword COVID-19 is kept in the keywords section as it will be routinely used to improve the citations.
- If possible, in introduction, the authors may cite recent prevalence or incidence data since the author have cited March 2023 data and it may be April 2023 for better understanding for the needs of this investigation.
Response: There is not a big change in mortality from March to May 2023 due to COVID_19 in However, we updated the number with the addition of a second digit after the decimal based on the latest mortality status.
- When referring to SPSS versions beginning from 19, authors should cite ‘IBM SPSS Statistics for Windows, version 25 (IBM Corp., Armonk, N.Y., USA)'.
Response: We have made the correction based on the direction of the respected reviewer.
- The authors should mention properly the word, “TNF-∝” instead of “TNF-”, all over the manuscript.
Response: Thanks for noticing this mistake, Corrected now in the manuscript.
- The conclusion seems very general. All conclusions must be convincing statements on what was found to be novel, impact based on the strong support of the data/results/discussion. Moreover, the authors may also be included the limitation of the present findings for a better understanding of the manuscript.
Response: We have expanded the conclusion. Limitations are included in the last paragraph of the discussion section.
Comments on the Quality of English Language
- The English need improvement since there are some grammatical and syntax errors in the manuscript. For example,
- in line number 114, the word “better” may be as “a better”;
- in line number 175, “Also it” as “Also, it”;
- in line number 176, “two ,or” as “two, or”;
- in line number 387, “Number” as “A number”.
Response: Thank you very much for your valuable suggestions. The whole manuscript is thoroughly checked for English language and corrected.

Reviewer 5 Report
The paper “Role of Natural Products in the Management of COVID-19: A Saudi Arabian Perspective” is generally well structured. I recommend this article to be published in the journal. Here are some suggestions:
1. Please consider changing the name from “COVID-19 virus” for “SARS-CoV-2”; “natural products (NP), natural products (NPs)” for “natural products (NPs)”; “NP” for “NPs”.
2. In lines 86-87, As new variants continue to evolve, it is important to discuss the drug resistance of different drug targets and the spectrum of antiviral activity. For example, Remdesivir drug resistance has been extensively studies. Please refer to: Nature Communications, 2022, 13, 1547; Antiviral Res. 2022, 198, 105247.
3. There is a lack of recent literature citations. For example, in lines 78-79, “The COVID-19 virus attracted the attention of scientists from all over the world who were hoping to discover and develop a treatment for the viral infection. (Viruses. 2022, 14, 443; Biomedicines. 2021, 9, 689)”.
4. For the benefits of the readers please list other agents for SARS-CoV-2 treatment. For example, Nirmatrelvir (Please refer to: Nature. 2023, 613(7944), 558-564.
5. Many natural products have been reported to have antiviral activity in cell culture, however, the mechanisms of action for many examples remain elusive. Please briefly discuss this issue.
6. In “Conclusions” section, it is better to include a paragraph describing the perspective of SARS-CoV-2 antiviral drug discovery. For example, what are the other promising drug targets? what are the desired properties for the next-generation of antiviral drugs based on natural products? In addition, combination therapy should be proposed as it is a common strategy for the treatment of viral infection. The quality of the Conclusions must be improved.
Minor editing of English language required
Author Response
Reviewer 5
Comments and Suggestions for Authors
The paper “Role of Natural Products in the Management of COVID-19: A Saudi Arabian Perspective” is generally well structured. I recommend this article to be published in the journal. Here are some suggestions:
- Please consider changing the name from “COVID-19 virus” for “SARS-CoV-2”; “natural products (NP), natural products (NPs)” for “natural products (NPs)”; “NP” for “NPs”.
Response: Thanks for your suggestions, correction is done throughout the manuscript as per the suggestion of the respected reviewer.
- In lines 86-87, As new variants continue to evolve, it is important to discuss the drug resistance of different drug targets and the spectrum of antiviral activity. For example, Remdesivir drug resistance has been extensively studies. Please refer to: Nature Communications, 2022, 13, 1547; Antiviral Res. 2022, 198, 105247.
Response: The following reference is now included in the introduction.
Gandhi, S., Klein, J., Robertson, A.J. et al. De novo emergence of a remdesivir resistance mutation during treatment of persistent SARS-CoV-2 infection in an immunocompromised patient: a case report. Nat Commun 13, 1547 (2022). https://doi.org/10.1038/s41467-022-29104-y
- There is a lack of recent literature citations. For example, in lines 78-79, “The COVID-19 virus attracted the attention of scientists from all over the world who were hoping to discover and develop a treatment for the viral infection. (Viruses. 2022, 14, 443; Biomedicines. 2021, 9, 689)”.
Response; We have included these two references now.
- For the benefits of the readers please list other agents for SARS-CoV-2 treatment. For example, Nirmatrelvir (Please refer to: Nature. 2023, 613(7944), 558-564.
Response: Thank you very much for your suggestion. Based on the recommendations of other four reviewers, we have included a number of references now to show the use of natural products in COVID-19. Therefore, we now have 58 references, kindly excuse me for this reference.
- Many natural products have been reported to have antiviral activity in cell culture, however, the mechanisms of action for many examples remain elusive. Please briefly discuss this issue.
Response: Included in the introduction section about the possible antiviral mechanism of natural products. Further we have discussed the mechanism of individual NPs in the discussion section.
- In “Conclusions” section, it is better to include a paragraph describing the perspective of SARS-CoV-2 antiviral drug discovery. For example, what are the other promising drug targets? what are the desired properties for the next-generation of antiviral drugs based on natural products? In addition, combination therapy should be proposed as it is a common strategy for the treatment of viral infection. The quality of the Conclusions must be improved.
Response: Included now. Thanks for your recommendation.
Comments on the Quality of English Language
Minor editing of English language required
Response: The whole manuscript is checked for English editing. Hope language has improved significantly.
